# A Conformal Frequency Reconfigurable Antenna with Multiband and Wideband Characteristics

**DOI:** 10.3390/s22072601

**Published:** 2022-03-29

**Authors:** Niamat Hussain, Adnan Ghaffar, Syeda Iffat Naqvi, Adnan Iftikhar, Dimitris E. Anagnostou, Huy H. Tran

**Affiliations:** 1Department of Smart Device Engineering, School of Intelligent Mechatronics Engineering, Sejong University, Seoul 05006, Korea; niamathussain@sejong.ac.kr; 2Department of Electrical and Electronics Engineering, Auckland University of Technology, Auckland 1010, New Zealand; aghaffar@aut.ac.nz; 3Telecommunication Engineering Department, University of Engineering & Technology Taxila (UET Taxila), Taxila 47050, Pakistan; iffat.naqvi@uettaxila.edu.pk; 4Department of Electrical and Computer Engineering, COMSATS University Islamabad, Islamabad 45550, Pakistan; adnaniftikhar@comsats.edu.pk; 5Institute of Signals, Sensors and Systems, Heriot Watt University, Edinburgh EH14 4AS, UK; d.anagnostou@hw.ac.uk; 6Faculty of Electrical and Electronic Engineering, PHENIKAA University, Hanoi 12116, Vietnam

**Keywords:** compact antenna, frequency reconfigurable, multiband, conformal antenna

## Abstract

A compact flexible multi-frequency antenna for smart portable and flexible devices is presented. The antenna consists of a coplanar waveguide-fed slotted circular patch connected to a rectangular secondary resonator (stub). A thin low-loss substrate is used for flexibility, and a rectangular stub in the feedline is deployed to attain wide operational bandwidth. A rectangular slot is etched in the middle of the circular patch, and a *p-i-n* diode is placed at its center. The frequency reconfigurability is achieved through switching the diode that distributes the current by changing the antenna’s electrical length. For the ON state, the antenna operates in the UWB region for −10 dB impedance bandwidth from 2.76 to 8.21 GHz. For the OFF state of the diode, the antenna operates at the ISM band (2.45/5.8 GHz), WLAN band (5.2 GHz), and lower X-band (8 GHz) with a minimum gain of 2.49 dBi and a maximum gain of 5.8 dBi at the 8 GHz band. Moreover, the antenna retains its performance in various bending conditions. The proposed antenna is suitable for modern miniaturized wireless electronic devices such as wearables, health monitoring sensors, mobile Internet devices, and laptops that operate at multiple frequency bands.

## 1. Introduction

The advancements in wireless technology and electromagnetic spectrum limitations have led to the development of multi-standard and multi-application devices. Considering this, an antenna with the characteristic of adaptability to various practical applications and standards is necessary. Hence, due to the dynamic characteristics and capability to modify properties such as polarization, radiation pattern, and frequency, along with system requirements, reconfigurable antennas have recently received a large amount of attention [1,2,3,4,5,6,7,8]. In particular, a frequency reconfigurable antenna is beneficial for various applications. Frequency reconfigurability can be achieved by using electrical switches [9,10,11], varactor diodes [12,13,14,15], *p-i-n* diodes [16,17,18,19], and radio-frequency micro-electromechanical systems (RF-MEMS) [20,21,22]. The electric switching technique has the advantage of lower voltage requirements, whereas RF-MEMS provide higher switching time. The *p-i-n* diodes have been widely used as reconfigurable techniques due to characteristics such as compactness and good switching time (1 to 100 microseconds) [23].

Moreover, the increasing requirement of compact, conformal devices, and wearable gadgets has drawn researchers’ attention towards flexible, low profile, and light-weight antennas. Thus, in addition to reconfigurability, flexibility is also a significant characteristic required by modern-day applications such as e-health monitoring, biosensing, and e-utility [24]. Although frequency reconfigurable antennas have been investigated widely in recent years, most of these designs have used rigid substrates. By comparison, few antenna designs have been recently reported in the literature, in which reconfigurability and flexibility have been integrated [25,26,27,28,29]. The antenna design proposed in [25] is an inkjet-printed flexible, reconfigurable antenna with overall dimensions of 53 × 31 mm^2^. In addition, a *p-i-n* diode is used for reconfigurability purposes. In another work [26], a reconfigurable antenna is reported for wearable applications. For this antenna, reconfigurability is realized using a *p-i-n* diode. In order to improve the performance of the antenna, an artificial magnetic conductor (AMC) surface is assimilated with the antenna. The proposed structure has an overall size of 83 × 89 mm^2^. Similarly, the work in [27] presented an inkjet-printed conformal antenna with geometrical dimensions of 30 × 40 mm^2^, where frequency reconfigurability is achieved by employing two diodes. Another reconfigurable antenna is reported in [28] with a substrate size of 59.8 × 59.8 mm^2^. In addition to reconfigurability, the antenna is flexible, and a conductive fabric is used to design the antenna on a polydimethylsiloxane (PDMS) substrate. Moreover, the antenna design demonstrated in [29] is a conformal antenna, and frequency reconfigurability is obtained using two *p-i-n* diodes. The geometrical size of the reported structure is 50 × 30 mm^2^. The works in [30,31] proposed flexible antennas with frequency reconfigurability. The proposed system presented in [30] obtains reconfigurability by employing two *p-i-n* diodes, and the dimensions of the substrate are 24 × 19 mm^2^ with a thickness of 1.53 mm. It can be observed that, although the reported antennas discussed here are both flexible and reconfigurable, these antennas have relatively larger dimensions, which restrict their usage in wearable and compact devices. In addition, these antennas have a relatively low gain. Thus, it can be deduced from the aforementioned discussion that a compact, high gain, flexible, and reconfigurable frequency antenna having a practical demonstration using diodes is still a challenge for researchers.

In order to overcome the limitations and discrepancies of the earlier reported antenna designs, this work proposes a compact and flexible antenna design for the ISM, WLAN, X-band, and UWB frequency bands. In addition, frequency reconfigurability is achieved by incorporating a *p-i-n* diode. Hence, the three bands can be merged into a single wideband (2.81–8.41 GHz) using this *p-i-n* diode. The omnidirectional radiation pattern and stable performance over a wide range of frequencies make this antenna desirable for a variety of applications.

## 2. Antenna Design Methodology

This section is divided into subsections for the ease of understanding.

### 2.1. Geometry of the Proposed Design

A circular planar radiator was chosen as the basis of the antenna design due to its inherent wide bandwidth. A coplanar waveguide (CPW) feed technique is used to excite a circular radiating patch that has a rectangular slot at its center. The circular patch has a radius *R* = 11 mm. The resonant frequency (*ƒ*) and the corresponding radius of the patch are estimated using [31]:(1)$$f=\frac{1.8412\;\cdot \;c}{4\pi R_{eƒƒ}\sqrt{\varepsilon_{eƒƒ}}}\;\;$$
where *c* is the free-space speed of light (3 × 10^8^ m/s), and $R_{eƒƒ}$ is the effective radius of the patch whose value is estimated using:(2)$$R_{eƒƒ}=R\sqrt{1+\frac{2H}{\pi \varepsilon_{eƒƒ}R}\left(ln\left(\frac{\pi R}{2H}\right)+1.7726\right)}\;$$

In (2), *R* is the physical radius of the patch, *H* is the thickness of the substrate and *ε_eff_* is the effective dielectric constant, which can be calculated by:(3)$$\varepsilon_{eff}=\frac{\varepsilon_{r}+1}{2}+\frac{\varepsilon_{r}-1}{2}\left\{{\left(1+12\frac{H}{A_{x}}\right)}^{-0.5}+0.04{\left(1-\frac{H}{A_{x}}\right)}^{2}\right\}\;$$
where $\varepsilon_{r}$ and $A_{x}$ are the dielectric constant and the width of the substrate, respectively.

The schematic of the proposed frequency reconfigurable antenna is shown in Figure 1. A rectangular stub with dimensions *Sx* × *Sy* is also inserted between the patch and feedline. The stub is employed to enhance the relatively narrow bandwidth of the circular patch. A rectangular slot is etched on the circular radiator to add additional capacitive load and allow the *p-i-n* diode to be inserted at the center of the slot to achieve frequency reconfigurability. The rectangular slot and the *p-i-n* diode control the amount of current by electrically connecting and disconnecting the upper part of the radiator with the lower part. With this arrangement, the diode generates the multiband and UWB modes. A detailed discussion on the working principle of the proposed antenna is presented in the subsequent sections.

The antenna is printed on a thin (*h* = 0.254 mm) Rogers RT5880LZ substrate with relative permittivity *ε_r_* = 2.1 and loss tangent *tanδ* = 0.0024 [32] This substrate is made of polytetrafluoroethylene (PTFE) composites and was chosen for its low-loss and flexible nature. These attributes make it a desirable substrate for flexible antennas.

### 2.2. Simulation Setup

The commercial finite element method-based HFSS [33] was used to simulate the proposed antenna. To avoid unwanted fabrication tolerance errors and effects from the SMA connector on the antenna, a 3D model of the 50-Ω SMA connector was designed and used to excite the proposed antenna in simulations. To model the real *p-i-n* diode switches, the equivalent circuit model of the Infineon model 3 BAR-50C-SC79 [34] was incorporated using lumped element parallel RLC boundary conditions. The equivalent circuit model of the diode in ON and OFF states is shown in Figure 2. In the ON state, a 4.5 Ω resistor is in series with a 0.15 nH inductor (Figure 2a). In the OFF state, a 0.15 nH inductor is in series with the parallel combination of a 5 kΩ resistor and a 0.15 pF capacitor (Figure 2b). Keeping in mind the practical issues related to measurements i.e., limiting the RF and DC current flowing towards the diode, an RF choke comprising a 68 nH inductor and a 1-kΩ resistor were utilized, as shown in Figure 2c. In addition, to accommodate biasing circuitry for the diode during measurements and eliminate any effect of the biasing circuitry on the radiating patch, two small biasing pads were incorporated on the bottom side of the proposed antenna. The optimized design is shown in Figure 1 and has the following dimensions: *A_Y_* = 35 mm, *A_x_* = 25 mm, *H* = 0.254 mm, *Cx* = 5 mm, *C_y_* = 11 mm, *S_x_* = 6 mm, *S_y_* = 9.5 mm, *r* = 11 mm, *F* = 1 mm, *G_3_* = 1 mm, *G_1_* = 1 mm, *G_2_* = 2.3 mm.

### 2.3. UWB Antenna Design

The fundamental CPW fed circular patch antenna was designed using (1)–(3). The resulting antenna resonates at 2.45 GHz with an impedance bandwidth of 2.25–2.72 GHz. Figure 3 (blue curve) illustrates the reflection coefficient of the elementary circular radiator. To broaden the narrowband operation, various techniques, including metamaterials, complex geometrical structures such as DGS, and etching slots, have been adopted [24,25,26,27]. Here, a uniplanar rectangular secondary resonator (stub) is introduced between the circular radiator and the feed. The stub acts as a high-frequency resonator and adds higher resonating frequencies to the circular radiator, resulting in a more wideband antenna. In other words, the field distribution of the conventional monopole antenna is altered due to the insertion of the rectangular stub, which supports multiple higher-order resonances instead of having only the single matched resonance offered by the circular radiator. The added bandwidth of the stub-loaded antenna results in a combined 5.8 GHz (2.4–3.8 GHz and 4.8–9.2 GHz) bandwidth, instead of only the 470 MHz (2.25 GHz–2.72 GHz) bandwidth that was achieved by the circular monopole itself.

Although a wideband antenna was achieved, a small portion of the bandwidth was slightly mismatched. Capacitor loading is a well-known technique to enable lower resonances and improve impedance matching [35]. Instead of loading the antenna with a physical capacitor, here, an electrical method was employed by etching a rectangular slot at the center of the circular radiator. The slot thickness was optimized so that the mismatched band could be matched by controlling the capacitance generated by the slot. The impedance bandwidth of the antenna before and after the insertion of the slot is shown in Figure 3 (black curve). It is observed that the stub-loaded antenna with the slot exhibits an ultrawide impedance bandwidth of 5.6 GHz, ranging from 2.81 GHz to 8.41 GHz.

### 2.4. Multiband Antenna Design

The designed UWB antenna was further utilized to develop a tri-band antenna. A small path is provided for the current to flow from the lower to the upper part of the radiator through a *p-i-n* diode in the ON state, as shown in Figure 3 (magenta curve). This path alters the capacitance of the slot, and the antenna exhibits a tri-band resonance. The first notch band was expected due to the disturbance in the capacitive load of the antenna. It is observed from Figure 3 that the antenna with the stub has a notched band inside the UWB region. The second notched band is due to the presence of two rectangular slots formed as a result of setting the diode ON. These slots behave like a band stop filter and thus cause higher band mitigation on the radiator. Moreover, a significantly lower current is present around the stub, which results in the suppression of the 3.75 and 6.5 GHz bands, as depicted in Figure 4b and Figure 4d, respectively. The geometric modifications in the resultant antenna exhibit three passbands having resonances at 2.45, 5.5, and 8 GHz, as shown in Figure 3 (magenta curve).

### 2.5. Parametric Analysis

A parametric study was performed to analyze the effects of the different antenna parameters on the antenna impedance. For better understanding, the diode OFF state was parametrically analyzed. It was noticed that the length (*Sy*) and width (*Sx*) of the rectangular patch deployed between the CPW and circular patch plays a key role in matching the impedance at different frequencies by controlling the amount of current flow on the antenna geometry. Figure 5a shows the effect of *Sy* on |S_11_|. An increase in the length of Sy from the optimized value of 9.5 mm results in better matching while disturbing the operational bandwidth. Conversely, a decrease in Sy results in comparatively better bandwidth. However, the reflection coefficient increases significantly.

Similarly, Figure 5b shows that a reduction in the length of *Sx* from the optimized value of 6 mm results in better matching (|S_11_| < −10 dB) at higher frequencies. Contrarily, by decreasing the width of the slot G_3_ from the optimized 1 mm, the reflection coefficient increases at lower frequencies, whereas an increase in G_3_ results in a mismatch at higher frequencies (Figure 5c). Considering these parameters, the optimized values were chosen to achieve the maximum bandwidth by considering |S_11_| < −10 dB.

## 3. Results and Discussion

The simulation results and its comparison with measured results are presented in this discussion.

### 3.1. Measurement Setup

To validate the working principle of the antenna, the antenna shown in Figure 1 was fabricated, and a photograph of the fabricated prototype is shown in Figure 6. Standard chemical etching was used for the fabrication and the scattering (S) parameters of the antenna were measured using a calibrated HP 8720D Vector Network Analyzer (VNA). To practically verify the reconfigurable operation, an Infineon (#BAR-50C SC79) *p-i-n* diode was soldered to the top side of the antenna, as depicted in Figure 6a. The biasing circuit was defined on the backside of the antenna (see Figure 6b) to prevent degradation of the radiation characteristics of the antenna. Two conducting vias were drilled in the radiating patches to provide bias voltages for the diode operation. A battery of 3 V was connected for the flow of current through the resistor and inductor, to turn on the diode, named ON-state. When the battery was disconnected, no current flowed through the diode and it behaved like an open circuit, referred to as its OFF-state.

### 3.2. Scattering Parameters

In Figure 7, the measured and simulated S-parameters are compared. When the diode is OFF, the antenna exhibits matching less than −10 dB over the 5.34 GHz band ranging from 2.76 to 8.1 GHz, compared with the simulated 5.6 GHz from 2.81 to 8.41 GHz. When the diode is switched ON, the antenna resonates at three frequencies: 2.47, 5.25, and 8.1 GHz, having an impedance bandwidth of 920 MHz (2.12–3.2 GHz), 2170 MHz (3.95–6.12 GHz), and 1200 MHz (7.71–8.83 GHz), respectively, as depicted in Figure 7b. The respective simulated values show resonances at 2.45, 5.2, and 8 GHz and with similar respective bandwidths.

### 3.3. Conformability Analysis

Flexibility and conformability are key requirements of flexible devices and a key advantage of the proposed antenna. Ideally, the antenna radiation should remain unchanged under both flat and flexed conditions. The conformability analysis was performed by bending the antenna on a cylindrical foam along the *x*- and *y*-axis, as depicted in Figure 8. The radius of the foam cylinder was chosen to be 20 mm as a realistic arm radius. With the diode in the OFF state, the antenna exhibits wide operational bandwidth and good agreement between simulations and measurements under both bending scenarios.

Similarly, Figure 9b illustrates that with the diode ON, the antenna exhibits a tri-band mode with almost identical |S_11_|. With the overall flat and flexed conditions having practically similar performance, the application of this antenna for both rigid and flexible/wearable devices was validated.

### 3.4. Far-Field Analysis

To observe the far-field radiation, the antenna was measured in a calibrated anechoic chamber as shown in Figure 10. The fabricated prototype was placed on a turntable in front of a broadband double-ridged horn at a far-field distance. Figure 11 shows the measured radiation patterns of the antenna for *p-i-n* diode ON and OFF at both *E*- and *H*-planes. The antenna has near-omnidirectional patterns at 2.45, 5.2, and 8 GHz in the principal *H*-plane, whereas for the *E*-plane, a tilted bi-directional pattern is observed, which is more prominent at higher frequencies (Figure 11a–c). A similar omni-directional *H*- plane pattern is observed for the ON state with a slightly tilted bi-directional *E*-plane at the selected frequencies of 3.2 and 5.8 GHz (Figure 12a,b). Overall, excellent agreement between measurements and simulations is observed at all frequencies for the diode’s ON and OFF states.

Figure 13a compares the simulated and measured gain in the ON and OFF states. The antenna exhibits a minimum gain of 2.49 dBi at 2.45 GHz and a maximum gain of 5.8 dBi at 8 GHz for the passband. Moreover, the gain decreases by up to −3 dBi in the band stop regions, which suffices to reject potential undesired interference in the ON state. Similarly, the simulated efficiency of the antenna has a minimum value of 80% in the operational band, whereas in the band stop region it decreases by up to 22%, as shown in Figure 13b. Thus, the antenna efficiently operates in the UWB and tri-band modes, depending upon the user requirements, by simply switching a single *p-i-n* diode.

### 3.5. Performance Comparision

The presented work was compared with the state-of-the-art works reported in literature, as summarized in Table 1. It can be observed that few works [17,18,27] are reconfigurable with good impedance bandwidth and significant gain; however, these designs have been employed on rigid substrates, which limit their effectiveness for various wireless communication applications. Other works presented in [25,26,28,29,30], demonstrate the advantages of reconfigurability and flexibility; however, these antenna structures either have larger dimensions or exhibit narrow bandwidth and low gain as compared to the proposed antenna. It is worth noting that this design uses only a single diode to achieve multiple reconfigurable bands. This comparative analysis verifies the usefulness and suitability of the proposed reconfigurable antenna for various modern-day wireless communication systems.

## 4. Conclusions

A compact and flexible CPW-fed antenna that consists of a circular patch connected to a rectangular stub is demonstrated here to operate at three different frequencies, or in the UWB region on demand, using frequency reconfigurability enabled by the use of only one *p-i-n* diode. The proposed antenna has the combined advantages of compact size, flexibility, and frequency reconfigurability with stable frequency and radiation pattern responses under planar and flexed conditions. The antenna operates in the ISM band at 2.45/5.8 GHz, the WLAN band at 5.2 GHz, and the lower X-band at 8 GHz with 2.49, 3.3, and 5.8 dBi gain, respectively. When the upper and lower parts of the circular patch are connected with a *p-i-n* diode, the three frequencies merge into a single wideband ranging from 2.76 to 8.41 GHz. The compactness, simple structure, and flexibility were studied by comparing the proposed antenna with current state-of-the-art designs. The stable performance of the proposed antenna in bending and flat conditions makes it an excellent candidate for compact wireless electronic devices simultaneously operating at different frequencies in the ISM, WLAN, and UWB bands such as mobile Internet devices, laptops, smartphones, health monitoring biosensors, and wearable electronics.

## Figures and Tables

**Figure 1 sensors-22-02601-f001:**
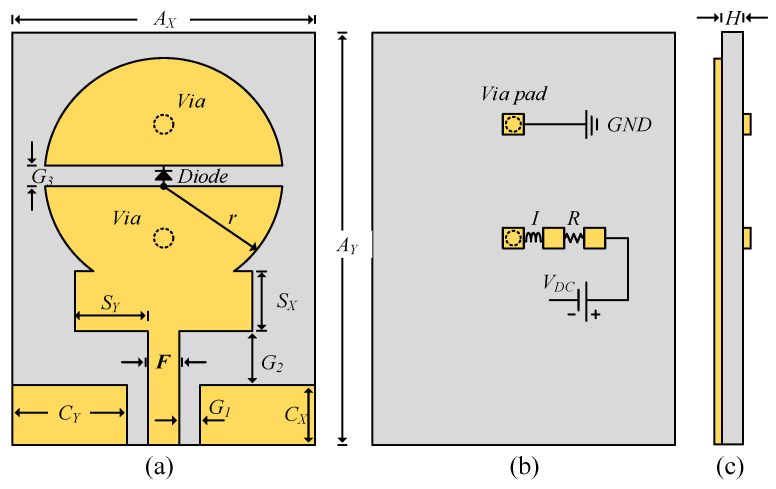
Schematic of proposed frequency reconfigurable antenna: (**a**) top-view, (**b**) bottom-view, and (**c**) side-view.

**Figure 2 sensors-22-02601-f002:**
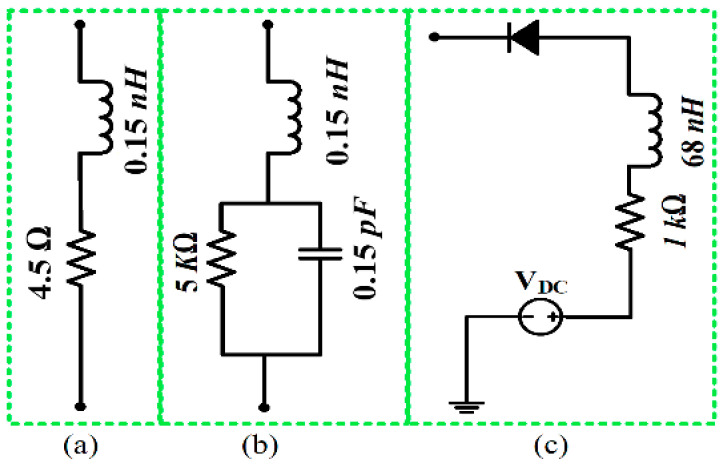
Diode equivalent model: (**a**) ON-state, (**b**) OFF-state, and (**c**) biasing circuit.

**Figure 3 sensors-22-02601-f003:**
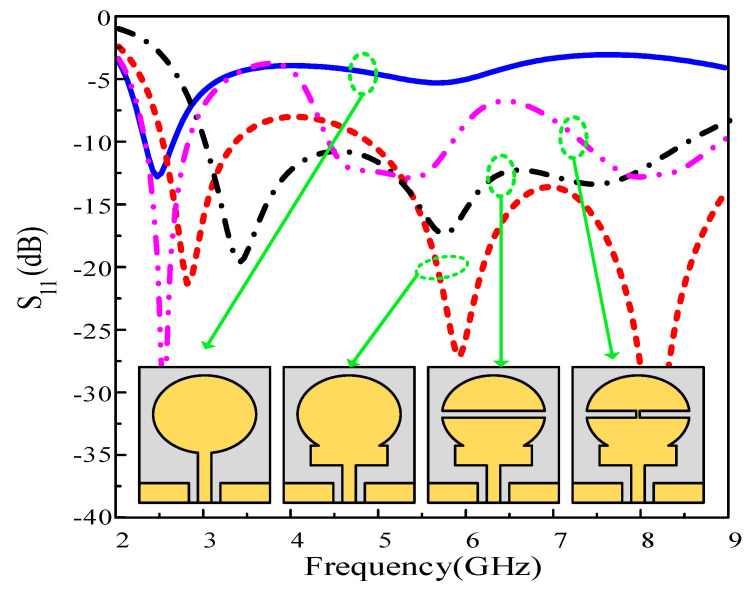
|S_11_| comparison among various steps included in the antenna design.

**Figure 4 sensors-22-02601-f004:**
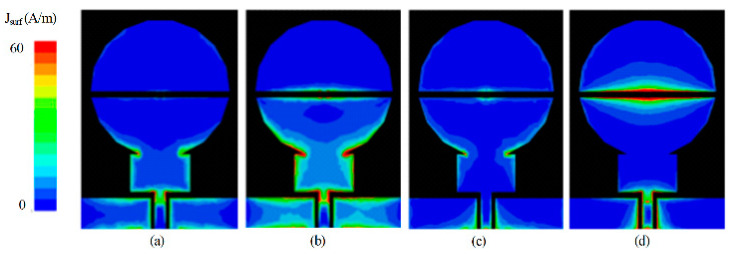
Distribution of current density on the surface of antenna at (**a**) 3.75 GHz [diode-OFF], (**b**) 3.75 GHz [diode-ON], (**c**) 6.5 GHz [diode-OFF], and (**d**) 6.5 GHz [diode-ON]. Comparison among the various steps included in the antenna design.

**Figure 5 sensors-22-02601-f005:**
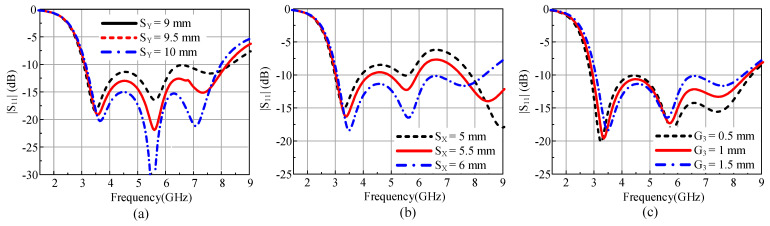
|S_11_| analysis for different (**a**) length of rectangular patch *Sy*, (**b**) width of rectangular patch *Sx*, and (**c**) width of slot *G*_3_.

**Figure 6 sensors-22-02601-f006:**
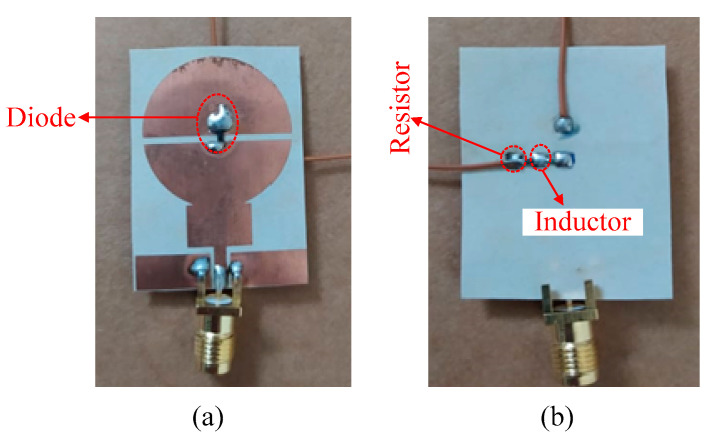
Fabricated prototype: (**a**) top-view and (**b**) bottom view.

**Figure 7 sensors-22-02601-f007:**
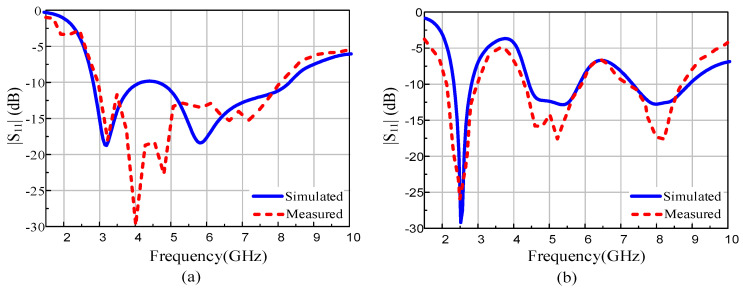
Simulated and measured |S_11_|: (**a**) diode ON and (**b**) diode OFF.

**Figure 8 sensors-22-02601-f008:**
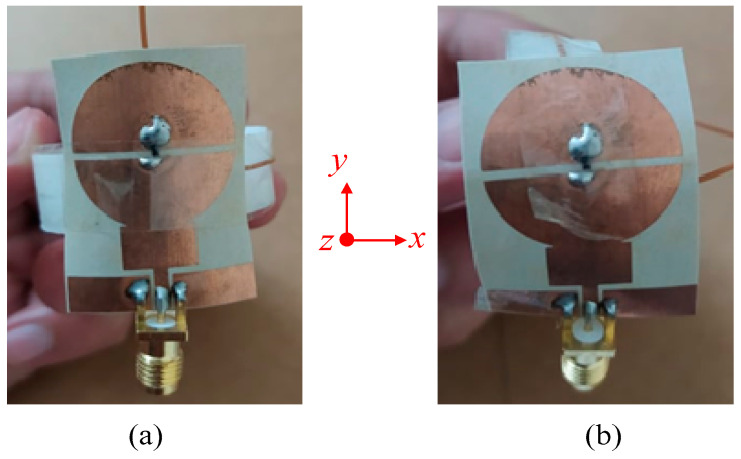
Antenna under conformal condition: (**a**) bending along the *x*-axis and (**b**) bending along the *y*-axis.

**Figure 9 sensors-22-02601-f009:**
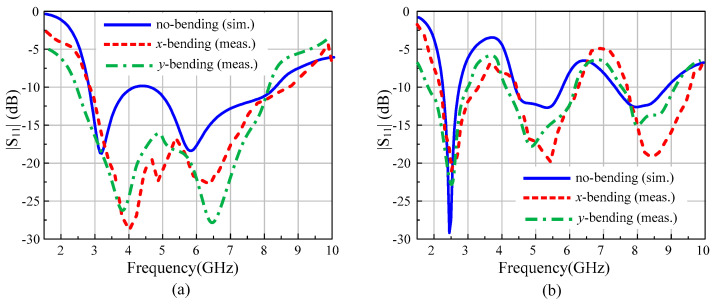
Conformability analysis of the proposed antenna: (**a**) diode ON, (**b**) diode OFF.

**Figure 10 sensors-22-02601-f010:**
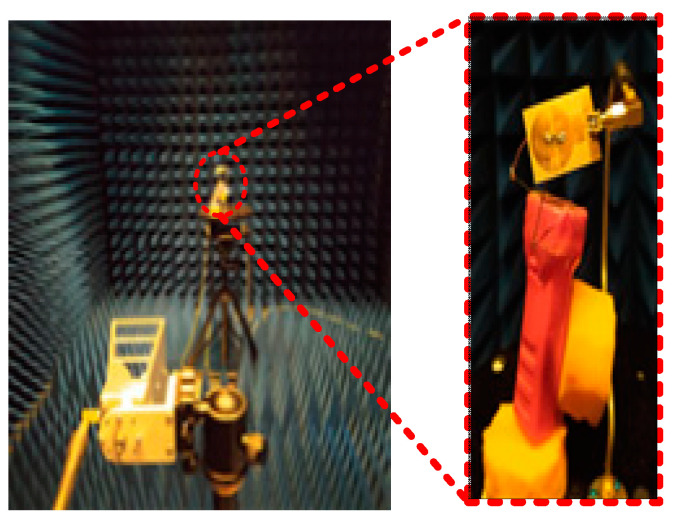
Far-field radiation pattern measurement setup.

**Figure 11 sensors-22-02601-f011:**
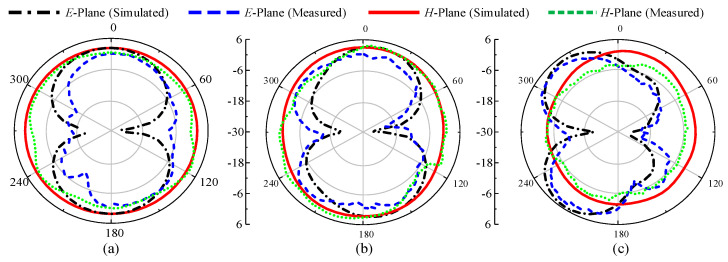
Radiation patterns of the proposed antenna for multiband mode: (**a**) 2.45 GHz, (**b**) 5.2 GHz, and (**c**) 8 GHz.

**Figure 12 sensors-22-02601-f012:**
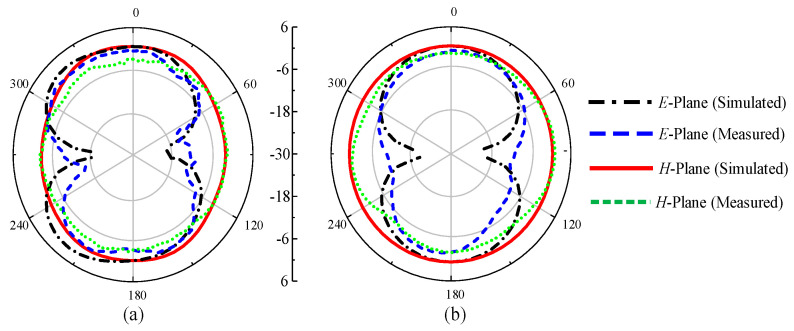
The proposed antenna’s radiation patterns for UWB mode: (**a**) 3.2 GHz and (**b**) 5.8 GHz.

**Figure 13 sensors-22-02601-f013:**
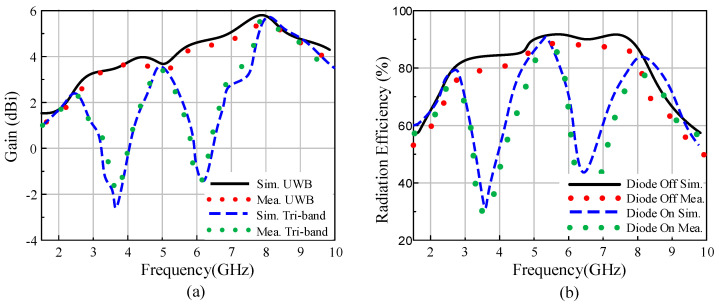
(**a**) Gain and (**b**) radiation efficiency of the proposed antenna.

**Table 1 sensors-22-02601-t001:** Performance comparison of the proposed antenna with other frequency reconfigurable antennas.

References	Antenna Size (mm^2^)	Operating Modes	No. of Diodes	Flexibility	Reconfigurable Frequency Range (GHz)	Peak Gain (dBi)
[17]	34.9 × 31	UWB and UWB with dual or tri narrow bands	4	No	2.13–10.5	5.2
[18]	88 × 83	UWB and tri-band	4	No	2–6	3.07
[25]	31 × 59	Single and dual band	1	Yes	2.27–3.77	1
[26]	89 × 83	Single and dual band	1	Yes	2.2–3.9	6.4
[27]	40 × 45	Single and dual band	2	No	1.5–4	2
[28]	51.8 × 59	Single band	2	Yes	2.36–3.9	3.6
[29]	50 × 33	Single and dual band	2	Yes	2.18–3.8	3.2
[30]	24 × 19	Single, dual and tri-band	2	Yes	2.3–5.75	3.73
This Work	35 × 25	UWB and tri-band	1	Yes	2.12–8.91	5.8

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
