# Peer review of "A Conformal Frequency Reconfigurable Antenna with Multiband and Wideband Characteristics"

_sensors, 2022, doi:10.3390/s22072601_

Round 1
Reviewer 1 Report
A frequency reconfigurable antenna was proposed. However, the innovation of the manuscript is limited. Compared to the published paper, the antenna structure presented in the article is an intermediate form, and there is no essential difference in the radiation mechanism. The reviewer thinks that the manuscript is not suitable for publish.
Author Response
Dear Reviewer,
Thanks for reviewing our manuscript.
The authors agree with the reviewer that the radiation mechanism of the antenna is not different from the other diode based reconfigurable antennas. However, the realization of multimode operation; UWB band and tri-band mode with the help of only one diode. Moreover, the antenna is flexible and retails its properties under bending conditions. Also, the performance comparison table shows the design merit of the antenna.
Please find the detailed response in the attached file.
Thank you.

Reviewer 2 Report
The paper presents a design of a compact flexible and reconfigurable patch antenna with multi band and wideband characteristics. The authors used a stub to enhance the antenna bandwidth and a thin substrate for flexibility. Moreover, the authors placed a p-i-n diode between the upper and lower parts of the radiator to achieve reconfigurability.
- The paper is readable, well written and coherence is well achieved.
- The simulation results are verified with the measured results, however, Not all results are well discussed and analyzed, see my comment below.
Overall, it is an important topic and should be considered for publication in Sensors once all comments have been addressed.
Additional comments:
- Introduction:
- Page 1, line 41, Avoid using run-on-expression such as “etc”
- Line 41-42, the authors claim that the P-i-n diode technique predominate other reconfigurable techniques due to higher switching time and lower biasing voltage, however, the optical switch (si) technique require lower voltage (e.g., 1.8 – 1.9 V) than the P-I-N diode technique which requires 3-5 V. Moreover, the RF MEMS technique has higher switching time (e.g., 1-200 Microseconds) than the p-i-n diode technique (i.e., 1-100 Microseconds). Also, it consumes more power than other techniques such as RF MEMS.
- Antenna design methodology
- Line 91, change the “ (3×108 m/s) “ to “ ().
- Lines 103 to 104, briefly explain how the activation of the p-i-n diode switches is achieved?
- Line 132, the variables for the antenna’s dimensions are not the same as the one on Figure 1. Also, Cx is not shown in Fig. 1. Please check and make sure that you are presenting the same variables in text and on Figure 1.
- Lines 150 – 151, the -10 dB bandwidth is from 5Ghz to 9.2 GHZ and not as claimed by the authors. From 3.5 GHz to 5 GHZ, the reflection coefficient is high (i.e., >-10 dB).
- Line 204, Typo in the title of the figure. Fig. 5 (a) Sy and not Sx while for Fig. 5 (b), it should be Sx and not Sy. Please fix it.
- Line 196, line 197 and Fig. 5 (c), Replace “G” by “G3”
- Line 193, replace “Return loss” by “reflection coefficient”. The correct definition of the Return Loss (RL) is: RL = 1/S11 (or RL_dB = -S11_dB); See the famous paper of Trevor Bird ( S. Bird, "Definition and Misuse of Return Loss [Report of the Transactions Editor-in-Chief]," in IEEE Antennas and Propagation Magazine, vol. 51, no. 2, pp. 166-167, April 2009, doi: 10.1109/MAP.2009.5162049.).
- 7 (a) the authors did not explain why there is disagreement between the measured and simulated S11 from 3.5 to 7 GHZ.
- In Table 2, the authors claimed that their proposed antenna achieved a BW ranging from 2.2 – 9.5 GHz while Figure 7 (a) shows that their proposed antenna achieved -10 dB bandwidth ranging from 2.7 – 8.3 GHz.
Author Response
Dear Reviewer,
We thank and appreciate for your valuable time in reviewing the manuscript and gave us constructive feedback. We have taken into consideration all the comments provided to us in the submitted manuscript. Please find the attached file for the corrections and additions that were made to the revised version. The revised manuscript is submitted in addition to this letter. All corrections highlighted in the revised manuscript for ease of tracking.
Best regards,
Authors

Round 2
Reviewer 2 Report
Thanks for addressing the comments!